# The Effect of *n*-3 PUFA Binding Phosphatidylglycerol on Metabolic Syndrome-Related Parameters and *n*-3 PUFA Accretion in Diabetic/Obese KK-*A^y^* Mice

**DOI:** 10.3390/nu11122866

**Published:** 2019-11-22

**Authors:** Liping Chen, Naoki Takatani, Fumiaki Beppu, Kazuo Miyashita, Masashi Hosokawa

**Affiliations:** Faculty of Fisheries Sciences, Hokkaido University, 3-1-1 Minato-cho, Hakodate, Hokkaido 041-8611, Japan; riheichin@eis.hokudai.ac.jp (L.C.); n-takatani@eis.hokudai.ac.jp (N.T.); fbeppu@fish.hokudai.ac.jp (F.B.); kmiya@fish.hokudai.ac.jp (K.M.)

**Keywords:** *n*-3 polyunsaturated fatty acid, phosphatidylglycerol, EPA, DHA, fatty acid accumulation, serum cholesterol, hepatic lipid

## Abstract

*n*-3 Polyunsaturated fatty acid binding phospholipids (*n*-3 PUFA-PLs) are known to be potent carriers of *n*-3 PUFAs and provide health benefits. We previously prepared *n*-3 PUFA binding phosphatidylglycerol (*n*-3 PUFA-PG) by phospholipase D-mediated transphosphatidylation. Because PG has excellent emulsifiability, *n*-3 PUFA-PG is expected to work as a functional molecule with properties of both PG and *n*-3 PUFAs. In the present study, the health benefits and tissue accretion of dietary *n*-3 PUFA-PG were examined in diabetic/obese KK-*A^y^* mice. After a feeding duration over 30 days, *n*-3 PUFA-PG significantly reduced the total and non-HDL cholesterols in the serum of diabetic/obese KK-*A^y^* mice. In the mice fed *n*-3 PUFA-PG, but not *n*-3 PUFA-TAG, hepatic lipid content was markedly alleviated depending on the neutral lipid reduction compared with the SoyPC-fed mice. Further, the *n*-3 PUFA-PG diet increased eicosapentaenoic acid and docosahexaenoic acid (DHA) and reduced arachidonic acid in the small intestine, liver, perirenal white adipose tissue, and brain, and the ratio of the n-6 PUFAs to *n*-3 PUFAs in those tissues became lower compared to the SoyPC-fed mice. Especially, the DHA level was more significantly elevated in the brains of *n*-3 PUFA-PG-fed mice compared to the SoyPC-fed mice, whereas *n*-3 PUFA-TAG did not significantly alter DHA in the brain. The present results indicate that *n*-3 PUFA-PG is a functional lipid for reducing serum and liver lipids and is able to supply *n*-3 PUFAs to KK-*A^y^* mice.

## 1. Introduction

Phosphatidylglycerol (PG) is a naturally occurring phospholipid (PL) with glycerol as a negatively charged polar head group. PG is found in bacteria as a major PL; in plants and mammals, PG is also detected as a minor PL with a varied fatty acid composition [1]. Furthermore, in several species of microalgae, *n*-3 polyunsaturated fatty acid (PUFA) binding PG (*n*-3 PUFA-PG) was contained, and was even found to be a major lipid component [2,3].

*n*-3 PUFAs are well known to have health benefits [4]. We usually intake *n*-3 PUFAs through the consumption of fish or fish oil to maintain adequate amounts of *n*-3 PUFAs in the body. Interestingly, the chemical forms of the lipids binding *n*-3 PUFAs have been suggested to affect the health benefit and bioavailability of *n*-3 PUFAs. Shirouchi et al. [5] described the superiority of *n*-3 PUFA-PC in prevention or alleviation obesity-related disorders through the suppression of fatty acid synthesis, enhancement of fatty acid beta-oxidation compared egg PC in Otsuka Long-Evans Tokushima fatty rats. Numerous studies have evaluated the bioavailability of several lipid forms, such as triacylglycerol (TAG), PL, ethyl esters, and monoacylglycerol (MAG) (mainly binding eicosapentaenoic acid (EPA) and docosahexaenoic acid (DHA)). DHA binding PL (DHA-PL) and DHA binding MAG (DHA-MAG) are suggested to be efficient carriers of dietary DHA in erythrocytes and plasma lipids when compared to DHA binding TAG (DHA-TAG) [6]. Liu et al. determined a higher efficacy of dietary DHA provided as PL than as TAG for brain DHA accretion in neonatal piglets [7]. Another prior study demonstrated that compared to fish oil TAG, roe-derived PL (*n*-3 PUFA containing PL (*n*-3 PUFA-PL)) administration enhanced lymphatic T DHA-PL absorption in unanesthetized rats, and plasma *n*-3 PUFAs are important as the supplying pool of *n*-3 PUFAs into various tissues [8]. However, in another study of mouse fed DHA-PL and DHA-TAG, DHA accretion in tissues did not differ between the administration of purified PL and TAG forms [9].

PG is a functional lipid with excellent emulsifiability and is expected to be applied in functional foods, cosmetics, and drug delivery. A previous study reported that palmitoyl-oleoyl-PG inhibits eicosanoid production in macrophages stimulated by *Mycoplasma pneumoniae*, insofar as saturated PG and saturated and unsaturated phosphatidylcholines did not have a significant effect on *M. pneumoniae*-induced arachidonic acid (ARA) release [10]. It has been demonstrated as well that PG-liposome decreases TNF-α production of lipopolysaccharide-stimulated macrophages in vitro [11]. On the other hand, we reported the preparation of *n*-3 PUFA-PG from salmon roe PL through phospholipase D (PLD)-mediated transphosphatidylation [12,13]. *n*-3 PUFA-PG has been given attention as a highly functional lipid with properties of both PG and *n*-3 PUFAs. We found that EPA and DHA at the s*n*-2 position of *n*-3 PUFA-PG were rapidly liberated by the pancreatic phospholipase A_2_ in vitro digestion model. However, there is little information about the beneficial health functions of dietary *n*-3 PUFA-PG and the incorporation and tissue accumulation of PUFAs from *n*-3 PUFA-PG. In the present study, we examined the effects of *n*-3 PUFA-PG on white adipose tissue (WAT) weight, blood glucose level, and serum and liver lipids of diabetic/obese KK-*A^y^* mice to explore its anti-obesity and anti-diabetic effects. Further, *n*-3 PUFA accretion in the tissues of KK-*A^y^* mice from dietary *n*-3 PUFA-PG was analyzed compared to *n*-3 PUFA-TAG and SoyPC diets.

## 2. Materials and Methods

### 2.1. Preparation of n-3 PUFA-PG

The crude PLs were obtained by the acetone precipitation of salmon roe lipids, as in a previous study [13]. The obtained insoluble precipitate (crude PLs) of 4.5 g was dissolved in 200 mL ethyl acetate and mixed with 200 mL of 0.2 M sodium acetate buffer (pH 5.6) containing 10 mM CaCl_2_; then, 75 g glycerol was added. Transphosphatidylation was initiated with the addition of 5 U PLD dissolved in a sodium acetate buffer, followed by stirring at 300 rpm at 37 °C for 24 h. The lipid components were extracted with chloroform/methanol/water (10:5:3, *v*/*v*/*v*), and the chloroform/methanol extracts underwent silica gel chromatography to separate their PG fractions. After elution of phosphatidylethanolamine (PE) with chloroform/methanol (8:2, *v*/*v*), PG was obtained with chloroform/methanol (7:3, *v*/*v*). We repeated the PLD-catalyzed transphosphatidylation to prepare sufficient amounts of *n*-3 PUFA-PG for the animal experiment. The purity of the *n*-3 PUFA-PG was greater than 95% (Appendix A), as confirmed by high performance chromatography as described in our previous paper [13].

### 2.2. Mouse Diet Composition

Diabetic/obese KK-*A^y^* mice (male, four-week old) were obtained from CLEA Japan, Inc., (Tokyo, Japan). The mice were housed individually and had free access to food and tap water. Room temperature and humidity were controlled at 23 ± 1 °C and 40% to 60% with a 12 h light/12 h dark cycle. The mice were acclimated to a modified AIN-93G diet with 7% soybean oil for one week and were then randomly assigned into three groups (*n* = 7 or 8) and fed the experimental diets for 30 days. Nutrients in the diets are shown in Appendix A. All animals were fed fresh diets every day. There were three experimental groups: The SoyPC group (5% soybean oil + 2% SoyPC), the *n*-3 PUFA-PG group (5% soybean oil + 2% *n*-3 PUFA-PG), and the *n*-3 PUFA-TAG (5.6568% soybean oil + 1.3432% fish oil). The SoyPC group and *n*-3 PUFA-PG group included diets with approximately the same calories for diabetic/obese KK-*A^y^* mice because the 2% soybean oil was substituted to PL, such as the SoyPC or *n*-3 PUFA-PG. The EPA and DHA in *n*-3 PUFA-PG and *n*-3 PUFA-TAG diets were adjusted to the approximately same amount to compare their bioavailabilities, as shown in Table 1. *n*-3 PUFA-TAG was prepared by mixing the EPA-28-TAG (Maruha Nichiro Corporation, Tokyo, Japan) and DHA-55-TAG (Maruha Nichiro Corporation, Tokyo, Japan) to adjust EPA and DHA contents to be the same as the *n*-3 PUFA-PG. The animal experiment was approved by the ethical committee at Hokkaido University, as well as all procedures for the use and care of animals. This research was carried out under the guidelines of the ethical committee of experimental animal care at Hokkaido University.

### 2.3. Sample Collection

Blood samples were collected from the tail vein of the mice after 6 h-fasting at 7, 14, 21, and 28 days during experimental feeding. Blood glucose was measured using a blood glucose monitor, a Glutest Neo Sensor (Sanwa Kagaku Kenkyusyo Co. Ltd., Aichi, Japan). After feeding with experimental diets for 30 days, the mice were sacrificed under ether anesthesia. Blood samples were collected from the caudal vena cava of the mice, and each sample of tissue was immediately excised, weighed, and stored at −30 °C. The serum lipid parameters of the KK-*A^y^* were analyzed by the Analytical Center of Hakodate Medical Association Japan.

### 2.4. Tissue Lipid Analysis

Total lipids (TLs) were extracted from the tissue by the Folch method [14]. Further, the TL from the liver (approx. 400 mg) were successively separated into neutral lipids (NLs) and PLs on a Sep-Pak^®^ silica cartridge (Waters, Dublin, Ireland) by elution with chloroform and methanol. The recovered lipids were checked by TLC (silica gel F254, Merck KGaA, Darmstadt, Germany), and developed in chloroform/methanol/H_2_O (65:25:4, *v*/*v*/*v*) solvent and sprayed with Dittmer reagent [15] to indicate PLs. Lipid content in the tissues was gravimetrically analyzed and was calculated as its per tissue weight. Cholesterol content in the TL was enzymatically measured using a commercial kit (Wako Chemical Industries, Ltd., Osaka, Japan). The fatty acid methyl esters (FAMEs) were prepared from the TL (approximately 10 mg) using a previous described method [16]. The FAMEs were injected into the gas chromatograph Shimazu GC-2014 gas chromatography system (Shimazu Corporation, Kyoto, Japan) equipped with a flame ionization detector with a fused silica capillary column, Omegawax 320 (0.32 mm × 30 m, Supelco, Bellefonte, PA, USA). The temperatures of the column, detector and injection port were set at 200, 260, and 250 °C, respectively. The amount of each fatty acid was calculated by using the internal standard (methyl tricosanoate, 23:0).

### 2.5. Statistical Analysis

Data are presented as the mean ± SE (*n* = 7 or 8). The difference between each diet group was analyzed by a one-way ANOVA with Turkey’s post hoc test using the SPSS V16.0 software (IBM Corp., New York, NY, USA). Differences were considered as significant at *p* < 0.05.

## 3. Results

### 3.1. Body and Tissue Weights, Food Intake and Water Intake

KK-*A^y^* mice were fed the experimental diets, with fatty acid compositions as shown in Table 1. The body weight gain and diet intake of the KK-*A^y^* mice were not different among the three groups after 30 days of feeding with experimental diets. No significant changes were found in the tissue weights between the three groups: SoyPC, *n*-3 PUFA-PG, and *n*-3 PUFA-TAG (Table 2).

### 3.2. Blood and Serum Parameters

The total cholesterol and non-HDL-cholesterol of KK-*A^y^* mice fed *n*-3 PUFA-PG were significantly lower than those of the mice fed SoyPC (Table 3). The *n*-3 PUFA-TAG supplementation also decreased total cholesterol and non-HDL cholesterol levels. Moreover, *n*-3 PUFA-PG did not significantly decrease HDL-cholesterol, while the HDL-cholesterol in the *n*-3 PUFA-TAG supplemented group decreased notably (Table 3). The blood glucose of the *n*-3 PUFA-PG-fed mice tended to be lower, but not significantly different, than that of the SoyPC-fed mice. AST and ALT were also no different among the three groups.

### 3.3. Fatty Acid Composition of Total Lipids from the Small Intestine

The EPA and DHA increased in the small intestine of mice fed with *n*-3 PUFA-PG or *n*-3 PUFA-TAG (Table 4). Neither the sum or individual saturated fatty acid (SFA) and monounsaturated fatty acid (MUFA) differed between the *n*-3 PUFA supplemented groups and the SoyPC group. The amount of EPA, *n*-3 docosapentaenoic acid (DPA*n*-3), and DHA in the small intestine were the at same levels in the *n*-3 PUFA-PG and *n*-3 PUFA-TAG groups. ARA content in the small intestine of the mice fed *n*-3 PUFA diets was significantly lower than that among the SoyPC-fed mice. The ratio of *n*-6/*n*-3 PUFAs was also significantly lower in the *n*-3 PUFA-PG and *n*-3 PUFA-TAG groups compared to the ratio of the SoyPC group. The EPA and DHA were incorporated into the small intestine largely via the replacement of ARA in the small intestine of mice fed *n*-3 PUFAs.

### 3.4. Lipid, Cholesterol and Fatty Acids in the Liver

The *n*-3 PUFA-PG group significantly alleviated the lipid accumulation in the liver compared to the SoyPC group (Table 5). The decrease in TL in the liver by *n*-3 PUFA-PG was dependent on NL reduction, but not PL content. A reduction by *n*-3 PUFA-PG feeding was significantly found for TL and NL (Table 5) in the liver of KK-*A^y^* mice. The *n*-3 PUFA-TAG did not significantly reduce hepatic TL and NL and significantly increased PL compared to SoyPC, though no significant difference in the TL, NL, and PL was found between the *n*-3 PUFA-PG and *n*-3 PUFA-TAG groups. On the other hand, there was no significance in the liver’s cholesterol contents among the three groups.

We next analyzed the fatty acid composition of hepatic lipids. Compared to the SoyPC group, the amount of EPA, DPA*n*-3, and DHA in hepatic TL, NL, and PL increased (Table 6, Table 7 and Table 8). However, their increase levels were not different between the *n*-3 PUFA-PG and *n*-3 PUFA-TAG groups. ARA was significantly less in the two *n*-3 PUFA administrated groups than that of the SoyPC group. The ratio of *n*-6/*n*-3 PUFAs in hepatic TL and NL also markedly decreased in both *n*-3 PUFA diets compared to the SoyPC diet (Table 6 and Table 7). In addition, both *n*-3 PUFA supplemented diets decreased 18:1*n*-9 levels in the liver. The decrease of 18:1*n*-9 by *n*-3 PUFA-PG was especially remarkable in the TL and NL in the liver. DHA is a major PUFA similar to linoleic acid (18:2*n*-6) in the hepatic PL of mice fed *n*-3 PUFA-PG or *n*-3 PUFA-TAG (Table 8). EPA and DHA from the *n*-3 PUFA-PG diet were predominately accreted into the PL in the liver.

### 3.5. Fatty Acid Composition of Brain and Perirenal WAT Lipids

The *n*-3 PUFA-PG diet significantly increased DHA levels, while the *n*-3 PUFA-TAG diet tended to increase DHA in the brain (Table 9). DPA*n*-3 was also increased and ARA was decreased more strongly by the *n*-3 PUFA-PG diet than the SoyPC diet. EPA accretion in the brain was very low in both *n*-3 PUFA diets. The levels of increase in the DHA and DPA*n*-3, and the decrease in ARA by the *n*-3 PUFA-PG diet were the almost same as those in the *n*-3 PUFA-TAG diet.

In the perirenal WAT, EPA, DPA*n*-3, and DHA levels were augmented by *n*-3 PUFA-PG or the *n*-3 PUFA-TAG (Table 10). However, the *n*-3 PUFAs content per tissue weight was comparatively lower in the perirenal WAT than in the liver and brain of the *n*-3 PUFA-fed mice. The amount of EPA and DHA did not vary between the *n*-3 PUFA-PG and the *n*-3 PUFA-TAG groups.

## 4. Discussion

The chemical structure of lipids has been reported to influence *n*-3 PUFA bioavailability [1]. It is well-documented that the *n*-3 PUFA-TAG is hydrolyzed into free fatty acids and s*n*-2-MAG by pancreatic lipase in the intestinal lumen [17]. After absorption of these hydrolysates, PL or TAG is re-synthesized in the small intestine epithelium and migrates into the lymph. On the other hand, PC is hydrolyzed into 1-acyl-lyso-PC and free fatty acids by pancreatic phospholipase A_2_ in the intestine [18]. A higher bioavailability of EPA and DHA from dietary *n*-3 PUFA-PL compared to *n*-3 PUFA-TAG has been reported using krill oil (EPA and DHA are mainly bound to PL) [8]. The previous research demonstrated, with high levels of confidence, that krill oil is effective for the management of hyperlipidemia at lower and equal doses and is significantly more effective than fish oil for the reduction of glucose, triglycerides, and LDL levels [19]. Another study indicates that krill oil is more effective than fish oil in increasing *n*-3 PUFA, thus reducing the *n*-6/*n*-3 PUFA ratio [20]. Krill oil with higher PL levels enhanced the bioavailability of *n*-3 PUFA compared to krill oil with lower PL levels [21]. Thus, the bioavailability of *n*-3 PUFAs has a close relationship with their esterified lipid forms.

In marine fish roe, *n*-3 PUFAs are mainly bound at the *sn*-2 position of PL [22]. Previously, we reported *n*-3 PUFA-PG preparation from salmon roe PL through PLD-mediated transphosphatidylation, and *n*-3 PUFAs binding to PG were predominately distributed at the s*n*-2 position of the glycerol backbone. PG is an essential biological component with important roles, such as cellular functions, in all eukaryotes and some prokaryotes [23]. In mammals, PG is a minor PL component of many intracellular membranes [24]. In keratinocytes, PG species containing PUFAs were effective at inhibiting rapidly proliferating keratinocytes, whereas PG species with MUFAs were effective at promoting proliferation in slowly dividing ones [25]. However, the health benefits of dietary *n*-3 PUFA-PG remain largely unknown. In addition, there is no information about the bioavailability of *n*-3 PUFAs from *n*-3 PUFA-PG after its digestion and absorption in body. For the utilization of *n*-3 PUFA-PG in functional foods and nutraceuticals, information regarding the health benefits and bioavailability of *n*-3 PUFAs released from *n*-3 PUFA-PG is important. Since obesity is recognized as worldwide problem because of risk factor for diabetes, hypertension and dyslipidemia to develop metabolic syndrome, we therefore examined the effect of *n*-3 PUFA-PG on WAT weight and blood glucose level, as well as the serum and liver lipid levels of type 2 diabetic/obesity model KK-*A^y^* mice to explore its preventive and alleviative activities. Further, the tissue accumulation of *n*-3 PUFAs was also analyzed for the first time in KK-*A^y^* mice by the supplementation of the *n*-3 PUFA-PG diet compared with an *n*-3 PUFA-TAG diet.

In this study, 2% *n*-3 PUFA-PG or n-PUFA-TAG with approximately equal amounts of EPA and DHA in the diets did not affect the body and WAT weight gain or the blood glucose level of obese/diabetic KK-*A^y^* mice compared to SoyPC after longitudinal (over 30 days) experimental feeding. To examine the anti-obesity effect of *n*-3 PUFA-PG, long-term feeding or more *n*-3 PUFA-PG in the diet might be required. On the other hand, 2% *n*-3 PUFA-PG in the diet significantly decreased the serum total cholesterol and non-HDL cholesterol as much as the *n*-3 PUFA-TAG. It is noteworthy that the *n*-3 PUFA-PG better alleviated lipid accumulation in the liver than SoyPC. The *n*-3 PUFA-TAG did not significantly reduce hepatic TL and NL compared to SoyPC, although no significant difference was found in the TL, NL, and PL between the *n*-3 PUFA-PG and *n*-3 PUFA-TAG groups. These data indicate that *n*-3 PUFA-PG exhibits serum cholesterol and hepatic lipid reduction in diabetic/obese KK-*A^y^* mice. The decreasing effect by *n*-3 PUFA-PG is suggested to have an identical or superior effect to the *n*-3 PUFA-TAG, and could be used as *n*-3 PUFAs source to improve the diabetic/obese associated lipid metabolism. To clarify the preventive effect of *n*-3 PUFA-PG on metabolic syndrome-related parameters, it is required for further investigation by using diet-induce obesity and diabetic mice and normal mice.

We further analyzed *n*-3 PUFA accumulation in several tissues of KK-*A^y^* mice fed an *n*-3-PUFA-PG diet. In the small intestine, liver and perirenal WAT, dietary *n*-3 PUFA-PG significantly elevated EPA, DPA*n*-3, and DHA, and reduced ARA at the same level as the *n*-3 PUFA-TAG diet. The *n*-3 PUFA-PG is a highly applicable lipid because of its excellent emulsifiability. In addition, several algal lipids, which have gained wide interest in various application in nutraceuticals, also contained *n*-3 PUFA-PG [26]. Therefore, the present study shows important results that *n*-3 PUFA-PG is an available dietary lipid source to supply *n*-3 PUFAs in the body.

DHA is one of the most abundant fatty acids in the brain to regulate important physiological functions [27]. The synthesis rate of DHA from ALA and EPA is very slow in an animal’s body. Therefore, the uptake of dietary DHA is necessary to maintain the essential levels [28]. Tracer studies indicate that phospholipid DHA targets the brain more effectively than DHA-TAG, although how this translates into higher brain DHA concentrations is not clearly understood [29]. Another study found that DHA esterified to PC, PE, or phosphatidylserine was more efficient at targeting the brain than DHA esterified to TAG in cortex and serum lipids [30]. In the present study, *n*-3 PUFA-PG increased DHA in the brain of diabetic/obese KK-*A^y^* mice. Interestingly, DHA accreted in the brain by *n*-3 PUFA-PG, but not by *n*-3 PUFA-TAG, was significantly higher than that of SoyPC. However, there was no significant difference between the *n*-3 PUFA-PG and *n*-3 PUFA-TAG groups, and the DHA content in the *n*-3 PUFA-PG diet was slightly higher than that in the *n*-3 PUFA-TAG diet. To clarify the *n*-3 PUFA-PG property needed to transport DHA in the brain, further investigation following a long feeding period and higher dose feeding is required.

## 5. Conclusions

Dietary *n*-3 PUFA-PG decreased the serum total cholesterol and non-HDL cholesterol of diabetic/obese KK-*A^y^* mice. In addition, *n*-3 PUFA-PG effectively alleviated lipid accumulation in the liver through the reduction of NL but not PL. In mice fed *n*-3 PUFA-PG, EPA, DPA*n*-3, and DHA were elevated, and ARA was reduced, in the small intestine, liver, perirenal WAT, and brain, and the ratio of n-6 PUFAs to *n*-3 PUFAs in the tissues was lower compared with the SoyPC fed mice. The present data indicate that *n*-3 PUFA-PG is a functional bioavailable lipid source for *n*-3 PUFAs and is at least the same or better than the *n*-3 PUFA-TAG.

## Figures and Tables

**Table 1 nutrients-11-02866-t001:** Major fatty acid composition of the experimental diets.

Fatty Acid (g/kg Diet)	SoyPC	*n*-3 PUFA-PG	*n*-3 PUFA-TAG
14:0	0.04	0.26	0.51
16:0	5.56	7.09	7.57
16:1*n*-7	0.02	0.45	0.37
18:0	2.17	3.00	2.74
18:1*n*-9	12.67	13.97	14.78
18:1*n*-7	0.84	1.19	1.13
18:2*n*-6	28.43	26.99	31.73
ALA (18:3*n*-3)	3.19	3.08	3.57
20:1*n*-9	0.14	0.38	0.31
ARA (20:4*n*-6)	ND	0.14	0.24
EPA (20:5*n*-3)	ND	2.14	2.28
DPA*n*-3 (22:5*n*-3)	ND	0.58	0.23
DHA (22:6*n*-3)	ND	2.63	2.49

SoyPC, soybean phosphatidylcholine; *n*-3 PUFA-PG, *n*-3 polyunsaturated fatty acid enriched phosphatidylglycerol; *n*-3 PUFA-TAG, *n*-3 polyunsaturated fatty acid enriched triacylglycerol; ND, not detected.

**Table 2 nutrients-11-02866-t002:** Body and tissue weights, food and water intake of KK-*A^y^* mice.

KK-*A^y^* Mice	SoyPC	*n*-3 PUFA-PG	*n*-3 PUFA-TAG
Initial BW (g)	25.81 ± 0.54	25.70 ± 0.50	25.99 ± 0.56
Final BW (g)	33.46 ± 0.81	33.73 ± 0.56	34.43 ± 0.76
BW Gain (g)	7.65 ± 0.45	8.03 ± 0.22	8.44 ± 0.37
Food Intake (g/day)	5.49 ± 0.15	5.60 ± 0.12	5.80 ± 0.13
Water Intake (g/day)	20.47 ± 1.25	20.17 ± 1.29	18.70 ± 1.38
Fece Excrement (dried g/day)	0.47 ± 0.02	0.42 ± 0.03	0.49 ± 0.02
Liver (g/100 g BW)	6.58 ± 0.16	6.06 ± 0.21	6.52 ± 0.15
Small Intestine (g/100 g BW)	3.20 ± 0.18	3.03 ± 0.18	2.73 ± 0.13
Brain (g/100 g BW)	1.14 ± 0.03	1.15 ± 0.04	1.14 ± 0.03
Kidney (g/100 g BW)	1.65 ± 0.05	1.64 ± 0.04	1.80 ± 0.08
Heart (g/100 g BW)	0.41 ± 0.01	0.43 ± 0.01	0.45 ± 0.02
Pancreas (g/100 g BW)	0.83 ± 0.08	0.71 ± 0.08	0.78 ± 0.08
Spleen (g/100 g BW)	0.24 ± 0.01	0.25 ± 0.01	0.27 ± 0.01
Muscle (g/100 g BW)	0.69 ± 0.01	0.72 ± 0.01	0.72 ± 0.02
BAT (g/100 g BW)	0.76 ± 0.05	0.73 ± 0.06	0.69 ± 0.05
Mesenteric WAT (g/100 g BW)	1.57 ± 0.05	1.32 ± 0.10	1.27 ± 0.06
Epididymal WAT (g/100 g BW)	4.03 ± 0.13	3.86 ± 0.12	3.95 ± 0.15
Perirenal WAT (g/100 g BW)	1.59 ± 0.09	1.41 ± 0.09	1.27 ± 0.04
Inguinal WAT (g/100 g BW)	3.06 ± 0.23	3.17 ± 0.19	2.75 ± 0.06

Data are expressed as the mean ± SE (*n* = 7 or 8). PUFA, polyunsaturated fatty acid; BW, body weight; BAT, brown adipose tissue; WAT, white adipose tissue.

**Table 3 nutrients-11-02866-t003:** Blood and serum parameters of KK-*A^y^* mice fed the experimental diets.

Blood and Serum Parameter	SoyPC	*n*-3 PUFA-PG	*n*-3 PUFA-TAG
AST (U/L)	82.75 ± 18.68	95.5 ± 28.62	82.71 ± 11.73
ALT (U/L)	39.88 ± 5.33	58.17 ± 26.23	40.28 ± 28.16
Total Cholesterol (mg/dL)	213.38 ± 9.06 ^a^	159.33 ± 7.33 ^b^	162.14 ± 10.75 ^b^
HDL-Cholesterol (mg/dL)	135.63 ± 3.11 ^a^	120.67 ± 4.48 ^ab^	116.71 ± 6.67 ^b^
Non-HDL-Cholesterol (mg/dL)	77.75 ± 7.00 ^a^	38.67 ± 3.16 ^b^	45.43 ± 4.30 ^b^
NL (mg/dL)	503.00 ± 56.24	508.5 ± 69.58	495.71 ± 68.30
Blood Glucose (mg/dL)	440 ± 31	333 ± 56	390 ± 36

Data are expressed as the mean ± SE (*n* = 7 or 8). The comparison was done by a one-way ANOVA with Turkey’s post hoc analysis. Different small letters denote significant differences at *p* < 0.05. AST, aspartate aminotransferase; ALT, alanine aminotransferase; HDL, high density lipoprotein; NL, neutral lipid.

**Table 4 nutrients-11-02866-t004:** Total lipid content and fatty acid composition of the small intestine in KK-*A^y^* mice fed experimental diets.

Fatty Acid (mg/g Small Intestine)	SoyPC	*n*-3 PUFA-PG	*n*-3 PUFA-TAG
SFA			
14:0	0.13 ± 0.04	0.16 ± 0.02	0.17 ± 0.02
16:0	4.24 ± 1.05	4.50 ± 0.44	4.35 ± 0.43
17:0	0.04 ± 0.01	0.04 ± 0.00	0.04 ± 0.00
18:0	1.10 ± 0.22	1.17 ± 0.07	1.10 ± 0.11
MUFA			
16:1*n*-7	0.65 ± 0.17	0.58 ± 0.08	0.52 ± 0.04
18:1*n*-9	6.01 ± 1.55	5.14 ± 0.59	4.90 ± 0.41
18:1*n*-7	0.33 ± 0.12	0.38 ± 0.03	0.33 ± 0.03
20:1*n*-9	0.12 ± 0.03	0.12 ± 0.01	0.11 ± 0.01
*n*-6 PUFA			
18:2*n*-6	6.75 ± 1.56	5.88 ± 0.48	5.81 ± 0.60
ARA (20:4*n*-6)	0.32 ± 0.06 ^a^	0.15 ± 0.02 ^b^	0.17 ± 0.02 ^b^
*n*-3 PUFA			
ALA (18:3*n*-3)	0.43 ± 0.10	0.40 ± 0.04	0.40 ± 0.05
EPA (20:5*n*-3)	ND	0.12 ± 0.01	0.12 ± 0.02
DPAn-3 (22:5*n*-3)	0.01 ± 0.01 ^b^	0.05 ± 0.00 ^a^	0.04 ± 0.01 ^a^
DHA (22:6*n*-3)	0.07 ± 0.01 ^b^	0.19 ± 0.02 ^a^	0.19 ± 0.02 ^a^
*n*-6/*n*-3 ratio	13.81 ± 0.25 ^a^	7.97 ± 0.36 ^b^	7.66 ± 1.14 ^b^
Total Lipid Content (mg/g Small Intestine)	6.46 ± 0.89	7.77 ± 0.71	8.22 ± 0.18

Data are expressed as the mean ± SE (*n* = 7 or 8). The comparison was done by a one-way ANOVA with Turkey’s post hoc analysis. Different small letters show significant differences at *p* < 0.05. SFA, saturated fatty acid; MUFA, monounsaturated fatty acid; PUFA, polyunsaturated fatty acid; ND, not detected.

**Table 5 nutrients-11-02866-t005:** Lipid and cholesterol contents in the liver of KK-*A^y^* mice fed experimental diets.

Content (mg/100 mg Liver)	SoyPC	*n*-3 PUFA-PG	*n*-3 PUFA-TAG
TL	6.69 ± 0.34 ^a^	5.25 ± 0.22 ^b^	6.06 ± 0.28 ^ab^
NL	3.45 ± 0.47 ^a^	2.22 ± 0.20 ^b^	3.08 ± 0.27 ^ab^
PL	2.07 ± 0.18 ^b^	2.46 ± 0.09 ^ab^	2.59 ± 0.06 ^a^
Cholesterol	1.77 ± 0.13	1.76 ± 0.17	1.87 ± 0.17

Values are represented as the mean ± SE (*n* = 7 or 8). The comparison was done by a one-way ANOVA with Turkey’s post hoc analysis. Different small letters across a row indicate significant differences at *p* < 0.05. PUFA, polyunsaturated fatty acid; TL, total lipid; NL, neutral lipid; PL, phospholipid.

**Table 6 nutrients-11-02866-t006:** Fatty acid composition of total lipid in the liver of KK-*A^y^* mice fed experimental diets.

Fatty Acid (mg/g Liver)	SoyPC	*n*-3 PUFA-PG	*n*-3 PUFA-TAG
SFA			
14:0	0.17 ± 0.03	0.10 ± 0.01	0.16 ± 0.03
16:0	13.16 ± 1.66	9.96 ± 0.75	13.78 ± 1.78
17:0	0.07 ± 0.01	0.07 ± 0.01	0.80 ± 0.73
18:0	4.24 ± 0.33	3.98 ± 0.23	4.26 ± 0.30
MUFA			
16:1*n*-7	1.31 ± 0.21	0.64 ± 0.04	1.18 ± 0.20
18:1*n*-9	16.64 ± 1.68 ^a^	8.01 ± 0.69 ^b^	13.93 ± 2.69 ^ab^
18:1*n*-7	1.69 ± 0.17 ^a^	0.63 ± 0.04 ^b^	0.54 ± 0.16 ^b^
20:1*n*-9	0.37 ± 0.04	0.33 ± 0.03	0.37 ± 0.03
*n*-6 PUFA			
18:2*n*-6	9.16 ± 0.82	7.18 ± 0.51	8.24 ± 0.56
ARA (20:4*n*-6)	3.48 ± 0.22 ^a^	1.71 ± 0.08 ^b^	2.03 ± 0.10 ^b^
*n*-3 PUFA			
ALA (18:3*n*-3)	0.10 ± 0.01 ^a^	0.05 ± 0.01 ^b^	0.07 ± 0.01 ^b^
EPA (20:5*n*-3)	0.13 ± 0.01 ^b^	1.01 ± 0.06 ^a^	0.99 ± 0.07 ^a^
DPAn-3 (22:5*n*-3)	0.11 ± 0.03 ^b^	0.39 ± 0.02 ^a^	0.43 ± 0.02 ^a^
DHA (22:6*n*-3)	1.70 ± 0.10 ^b^	3.22 ± 0.16 ^a^	3.11 ± 0.10 ^a^
*n*-6/*n*-3 ratio	6.48 ± 0.22 ^a^	1.93 ± 0.09 ^b^	2.27 ± 0.11 ^b^

Values are represented as the mean ± SE (*n* = 7 or 8). The comparison was done by a one-way ANOVA with Turkey’s post hoc analysis. Different small letters show significant differences at *p* < 0.05. SFA, saturated fatty acid; MUFA, monounsaturated fatty acid; PUFA, polyunsaturated fatty acid.

**Table 7 nutrients-11-02866-t007:** Fatty acid composition of the neutral lipid in the liver of KK-*A^y^* mice fed experimental diets.

Fatty Acid (mg/g Liver)	SoyPC	*n*-3 PUFA-PG	*n*-3 PUFA-TAG
SFA			
14:0	0.18 ± 0.02	0.12 ± 0.03	0.16 ± 0.03
16:0	8.54 ± 0.69	5.98 ± 0.76	8.05 ± 0.99
17:0	0.03 ± 0.01	0.03 ± 0.00	0.03 ± 0.01
18:0	0.68 ± 0.07	0.56 ± 0.08	0.69 ± 0.11
21:0	0.08 ± 0.01	0.05 ± 0.00	0.06 ± 0.01
MUFA			
16:1*n*-7	1.14 ± 0.08 ^a^	0.63 ± 0.09 ^b^	0.86 ± 0.12 ^ab^
17:1*n*-9	0.06 ± 0.00	0.05 ± 0.01	0.05 ± 0.01
18:1*n*-9	14.20 ± 1.73 ^a^	7.55 ± 0.92 ^b^	11.06 ± 1.34 ^ab^
18:1*n*-7	0.69 ± 0.20 ^a^	0.48 ± 0.05 ^b^	0.67 ± 0.09 ^a^
20:1*n*-9	0.35 ± 0.03 ^a^	0.20 ± 0.02 ^b^	0.29 ± 0.03 ^ab^
*n*-6 PUFA			
18:2*n*-6	5.87 ± 0.54	4.90 ± 0.64	6.20 ± 1.04
ARA (20:4*n*-6)	0.18 ± 0.03 ^a^	0.09 ± 0.01 ^b^	0.13 ± 0.02 ^b^
*n*-3 PUFA			
ALA (18:3*n*-3)	0.33 ± 0.04	0.34 ± 0.05	0.42 ± 0.07
EPA (20:5*n*-3)	0.01 ± 0.00 ^b^	0.21 ± 0.02 ^a^	0.28 ± 0.03 ^a^
DPAn-3 (22:5*n*-3)	0.02 ± 0.01 ^b^	0.17 ± 0.02 ^a^	0.21 ± 0.03 ^a^
DHA (22:6*n*-3)	0.10 ± 0.02 ^b^	0.48 ± 0.05 ^a^	0.65 ± 0.07 ^a^
*n*-6/*n*-3 ratio	13.65 ± 0.88 ^a^	3.96 ± 0.27 ^b^	4.05 ± 0.31 ^b^

Data are expressed as the mean ± SE (*n* = 7 or 8). The comparison was done by a one-way ANOVA with Turkey’s post hoc analysis. Different small letters indicate significant differences at *p* < 0.05. Means without common letters within the same row differ significantly from the SoyPC group. SFA, saturated fatty acid; MUFA, monounsaturated fatty acid; PUFA, polyunsaturated fatty acid.

**Table 8 nutrients-11-02866-t008:** Fatty acid composition of phospholipids in the liver in KK-*A^y^* mice fed experimental diets.

Fatty Acid (mg/g Liver)	SoyPC	*n*-3 PUFA-PG	*n*-3 PUFA-TAG
SFA			
14:0	0.01 ± 0.00	0.01 ± 0.00	0.01 ± 0.00
16:0	3.13 ± 0.25	4.44 ± 0.38	4.53 ± 0.64
17:0	0.03 ± 0.00	0.04 ± 0.00	0.04 ± 0.00
18:0	3.11 ± 0.23	3.46 ± 0.21	3.76 ± 0.47
21:0	0.25 ± 0.02	0.20 ± 0.01	0.22 ± 0.03
MUFA			
16:1*n*-7	0.11 ± 0.01	0.13 ± 0.02	0.12 ± 0.02
17:1*n*-9	0.01 ± 0.00	0.01 ± 0.00	0.01 ± 0.00
18:1*n*-9	1.07 ± 0.07	1.18 ± 0.09	1.26 ± 0.17
18:1*n*-7	0.29 ± 0.02 ^a^	0.19 ± 0.01 ^b^	0.21 ± 0.03 ^b^
20:1*n*-9	0.08 ± 0.03	0.04 ± 0.00	0.05 ± 0.01
*n*-6 PUFA			
18:2*n*-6	2.46 ± 0.21	2.83 ± 0.19	3.00 ± 0.39
ARA (20:4*n*-6)	2.76 ± 0.20 ^a^	1.45 ± 0.10 ^b^	1.80 ± 0.17 ^b^
*n*-3 PUFA			
ALA (18:3*n*-3)	0.02 ± 0.00 ^b^	0.04 ± 0.00 ^a^	0.03 ± 0.00 ^a^
EPA (20:5*n*-3)	0.08 ± 0.00 ^b^	0.66 ± 0.04 ^a^	0.63 ± 0.09 ^a^
DPAn-3 (22:5*n*-3)	0.08 ± 0.01 ^b^	0.19 ± 0.01 ^a^	0.17 ± 0.02 ^a^
DHA (22:6*n*-3)	1.33 ± 0.07 ^b^	2.34 ± 0.12 ^a^	2.38 ± 0.21 ^a^
*n*-6/*n*-3 ratio	3.43 ± 0.09 ^a^	1.32 ± 0.05 ^b^	1.49 ± 0.05 ^b^

Data are the mean ± SE (*n* = 7 or 8). The comparison was done by one-way ANOVA with Turkey’s post hoc analysis. Different small letters denote significant differences at *p* < 0.05. SFA, saturated fatty acid; MUFA, monounsaturated fatty acid; PUFA, polyunsaturated fatty acid.

**Table 9 nutrients-11-02866-t009:** Total lipid content and fatty acid composition of the brain in KK-*A^y^* mice fed experimental diets.

Fatty Acid (mg/g Brain)	SoyPC	*n*-3 PUFA-PG	*n*-3 PUFA-TAG
SFA			
14:0	0.04 ± 0.00	0.04 ± 0.00	0.04 ± 0.00
16:0	6.43 ± 0.38	6.94 ± 0.27	6.50 ± 0.32
17:0	0.06 ± 0.00	0.07 ± 0.00	0.06 ± 0.00
18:0	6.26 ± 0.27	6.60 ± 0.19	6.25 ± 0.25
21:0	0.17 ± 0.01	0.22 ± 0.01	0.19 ± 0.00
MUFA			
16:1*n*-7	0.19 ± 0.01	0.21 ± 0.01	0.20 ± 0.01
17:1*n*-9	0.02 ± 0.00	0.03 ± 0.00	0.02 ± 0.00
18:1*n*-9	5.15 ± 0.21	5.61 ± 0.19	5.21 ± 0.19
18:1*n*-7	1.19 ± 0.05	1.20 ± 0.04	1.08 ± 0.04
20:1*n*-9	0.67 ± 0.02	0.71 ± 0.02	0.63 ± 0.02
*n*-6 PUFA			
18:2*n*-6	0.39 ± 0.02	0.42 ± 0.04	0.41 ± 0.05
ARA (20:4*n*-6)	2.93 ± 0.10 ^a^	2.62 ± 0.07 ^b^	2.55 ± 0.09 ^b^
*n*-3 PUFA			
ALA (18:3*n*-3)	2.93 ± 0.10	2.62 ± 0.07	2.55 ± 0.09
EPA (20:5*n*-3)	ND	0.04 ± 0.00	0.03 ± 0.00
DPAn-3 (22:5*n*-3)	0.03 ± 0.01 ^b^	0.12 ± 0.00 ^a^	0.10 ± 0.00 ^a^
DHA (22:6*n*-3)	4.92 ± 0.14 ^b^	5.43 ± 0.09 ^a^	5.30 ± 0.00 ^ab^
*n*-6/*n*-3 ratio	0.67 ± 0.01 ^a^	0.54 ± 0.01 ^b^	0.54 ± 0.01 ^b^
Total Lipid Content (mg/g Brain)	7.86 ± 0.15	7.96 ± 0.09	7.48 ± 0.22

Data are the mean ± SE (*n* = 7 or 8). The comparison was done by a one-way ANOVA with Turkey’s post hoc analysis. Different small letters indicate significant differences at *p* < 0.05. SFA, saturated fatty acid; MUFA, monounsaturated fatty acid; PUFA, polyunsaturated fatty acid; ND, not detected.

**Table 10 nutrients-11-02866-t010:** Total lipid content and fatty acid composition of the perirenal WAT in KK-*A^y^* mice fed experimental diets.

Fatty Acid (mg/g Perirenal WAT)	SoyPC	*n*-3 PUFA-PG	*n*-3 PUFA-TAG
SFA			
14:0	1.65 ± 0.13	1.65 ± 0.17	1.92 ± 0.16
16:0	32.02 ± 1.56	30.95 ± 2.83	33.93 ± 2.58
17:0	0.20 ± 0.01	0.24 ± 0.02	0.23 ± 0.02
18:0	3.08 ± 0.21	3.80 ± 0.30	3.57 ± 0.27
21:0	0.14 ± 0.01	0.09 ± 0.01	0.12 ± 0.01
MUFA			
16:1*n*-7	8.39 ± 0.52	6.50 ± 0.63	6.92 ± 0.58
17:1*n*-9	0.26 ± 0.01	0.27 ± 0.03	0.27 ± 0.02
18:1*n*-9	53.28 ± 2.66	42.47 ± 3.11	50.63 ± 3.69
18:1*n*-7	ND	1.91 ± 0.52	ND
20:1*n*-9	0.96 ± 0.04 ^ab^	0.92 ± 0.06 ^b^	1.04 ± 0.08 ^a^
*n*-6 PUFA			
18:2*n*-6	51.92 ± 2.11	43.24 ± 3.07	49.42 ±4.12
ARA (20:4*n*-6)	0.24 ± 0.02 ^a^	0.12 ± 0.01 ^b^	0.16 ± 0.02 ^b^
*n*-3 PUFA			
ALA (18:3*n*-3)	3.48 ± 0.15	3.05 ± 0.22	3.44 ± 0.31
EPA (20:5*n*-3)	ND	0.23 ± 0.02	0.22 ± 0.03
DPAn-3 (22:5*n*-3)	ND	0.12 ± 0.01 ^a^	0.05 ± 0.02 ^b^
DHA (22:6*n*-3)	ND	0.26 ± 0.03	0.20 ± 0.04
*n*-6/*n*-3 ratio	15.01 ± 0.33 ^a^	11.87 ± 0.31 ^b^	12.82 ± 0.50 ^b^
Total Lipid Content (mg/g Perirenal WAT)	83.94 ± 1.97	85.48 ± 2.65	86.27 ± 2.87

Data are the mean ± SE (*n* = 7 or 8). The comparison was done by a one-way ANOVA with Turkey’s post hoc analysis. Different small letters indicate significant differences at *p* < 0.05. SFA, saturated fatty acid; MUFA, monounsaturated fatty acid; PUFA, polyunsaturated fatty acid; ND, not detected.

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
