# Peer review of "The Effect of *n*-3 PUFA Binding Phosphatidylglycerol on Metabolic Syndrome-Related Parameters and *n*-3 PUFA Accretion in Diabetic/Obese KK-*A^y^* Mice"

_nutrients, 2019, doi:10.3390/nu11122866_

Round 1
Reviewer 1 Report
If the aim is to study the bioavailability of different n-3 PUFA forms in animals, why use obese/diabetic KK-Ay mice? The percentage of n-3 PUFA-PG and n-3 PUFA-TAG in the diet are different, PG-form is 2%, but TAG-form is 1.3432%, please explain it. The description of the function of PG on keratinocyte is not suitable for the background. Please provide the HPLC data of n-3 PUFA PG. What’s the novelty and values of the n-3 PUFA PG? If the mice increase the intake of 2.5-3% TAG form, maybe the effects of PG (2%) are similar to the n-3 PUFA-TAG.Author Response
Reviewer 1
If the aim is to study the bioavailability of different n-3 PUFA forms in animals, why use obese/diabetic KK-Ay mice?
Reply: Thank you for your comment. As in title, the purpose of our research is to investigate effect of n-3 PUFA binding phosphatidylglycerol on metabolic syndrome-related parameters and n-3 PUFA accretion in diabetic/obese mice. So, in this study, we used KK-Ay as type 2 diabetic/obese model mice. To explain those points, we revised as below.
P2, L23-25; “In the present study, we examined the effects of n-3 PUFA-PG on white adipose tissue weight, blood glucose level, and serum and liver lipids of diabetic/obese KK-Ay mice to explore its anti-obesity and anti-diabetic effects.”
P10, L14-18; “Since obesity is recognized as worldwide problem because of risk factor for diabetes, hypertension and dyslipidemia to develop metabolic syndrome, we therefore examined the effect of n-3 PUFA-PG on white adipose tissue’s weight and blood glucose level, as well as the serum and liver lipid levels of type 2 diabetic/obesity model KK-Ay mice to explore its preventive and alleviative activities.”
The percentage of n-3 PUFA-PG and n-3 PUFA-TAG in the diet are different, PG-form is 2%, but TAG-form is 1.3432%, please explain it.
Reply: We adjusted EPA and DHA contents in the n-3 PUFA-PG and n-3 PUFA-TAG diets to compare their accumulation in the tissues. So the percentage of n-3 PUFA-PG and n-3 PUFA-TAG in the diets are different. We explained the reason as below.
P3, L3-4; “The EPA and DHA in n-3 PUFA-PG and n-3 PUFA-TAG diets were adjusted to the approximately same amount as shown in Table 1 to compare their accumulation in tissues.”
The description of the function of PG on keratinocyte is not suitable for the background.
Reply: P2; Thank you for your comment. We deleted the function of PG on keratinocytes.
Please provide the HPLC data of n-3 PUFA PG.
Reply: Sorry for insufficient notation. We cited our previous paper describing HPLC conditions of n-3 PUFA-PG as Ref [13] in “2.1. Preparation of n-3 PUFA-PG” and show HPLC chromatogram as supplemental Figure 1.
What’s the novelty and values of the n-3 PUFA PG?
Reply: Thank you for your important comment. We described novel points of our study as below.
P10, L14-20; “Since obesity is recognized as worldwide problem because of risk factor for diabetes, hypertension and dyslipidemia to develop metabolic syndrome, we therefore examined the effect of n-3 PUFA-PG on white adipose tissue’s weight and blood glucose level, as well as the serum and liver lipid levels of type 2 diabetic/obesity model KK-Ay mice to explore its preventive and alleviative activities. Further, the tissue accumulation of n-3 PUFAs was also analyzed for the first time in KK-Ay mice by the supplementation of the n-3 PUFA-PG diet compared with an n-3 PUFA-TAG diet.”
If the mice increase the intake of 2.5-3% TAG form, maybe the effects of PG (2%) are similar to the n-3 PUFA-TAG.
Reply: We think n-3 PUFA-PG is a multifunctional lipid with both properties of PG and n-3 PUFAs. This is very important and novel points of the present study. To explain those points, we added explanation “The n-3-PUFA-PG is a highly applicable lipid because of excellent emulsifiability. In addition, several algal lipids, which have gained wide interest in various application in nutraceuticals, also contained n-3 PUFA-PG [26]. Therefore, the present study shows important results that n-3 PUFA-PG is an available dietary lipid source to supply n-3 PUFAs in the body.” at P10, Line 38-42.

Reviewer 2 Report
This manuscript studied the effect of n-3 PUFA-PG on metabolic syndrome related parameters and fatty acid content in different tissues in diabetic/obese mice. I have a few general questions or comments.
1. Why diabetic/obese KK-Ay mice was chosen for the study should be explained in the introduction? Will n-3 PUFA-PG have similar effect on healthy mice?
2. The DPAn-3 content in n-3PUFA-PG diet is higher than that of n-3 PUFA-TAG diet, will this have any influences on the results?
3. For the lipid class analyses and cholesterol analyses, no references were cited. How accurate is the method as compared to traditional HPLC or HPTLC and GC?
4. It would be better to provide the sum of the fatty acid, or total lipid content in the tables showing fatty acid composition, to have an idea whether they differ in the total amount.
Author Response
Reviewer 2
This manuscript studied the effect of n-3 PUFA-PG on metabolic syndrome related parameters and fatty acid content in different tissues in diabetic/obese mice. I have a few general questions or comments.
Why diabetic/obese KK-Ay mice was chosen for the study should be explained in the introduction? Will n-3 PUFA-PG have similar effect on healthy mice?
Reply: Thank you for your suggestion and comment. As in title, the purpose of our research is to investigate health benefits of n-3 PUFA binding phosphatidylglycerol on metabolic syndrome-related parameters and n-3 PUFA accretion in diabetic/obese mice. So, in this study, we used obese/type 2 diabetic model KK-Ay as mice. As your comment, it is also important to estimate health benefits by using diet-induced obesity and diabetic mice and normal mice. It is required further investigation. To explain why we used diabetic/obese KK-Ay in the present study, we described as below.
P2, L22-24; “In the present study, we examined the effects of n-3 PUFA-PG on white adipose tissue weight, blood glucose level, and serum and liver lipids of diabetic/obese KK-Ay mice to explore its anti-obesity and anti-diabetic effects.”
P10, L14-20; “Since obesity is recognized as worldwide problem because of risk factor for diabetes, hypertension and dyslipidemia to develop metabolic syndrome, we therefore examined the effect of n-3 PUFA-PG on white adipose tissue’s weight and blood glucose level, as well as the serum and liver lipid levels of type 2 diabetic/obesity model KK-Ay mice to explore its preventive and alleviative activities. Further, the tissue accumulation of n-3 PUFAs was also analyzed for the first time in KK-Ay mice by the supplementation of the n-3 PUFA-PG diet compared with an n-3 PUFA-TAG diet.”
P10, L31-35; “The decreasing effect by n-3 PUFA-PG is suggested to have an identical or superior effect to the n-3 PUFA-TAG, and could be used as n-3 PUFAs source to improve the diabetic/obese associated lipid metabolism. To clarify the preventive effect of n-3 PUFA-PG on metabolic syndrome-related parameters, it is required for further investigation by using diet-induce obesity and diabetic mice and normal mice.”
The DPAn-3 content in n-3PUFA-PG diet is higher than that of n-3 PUFA-TAG diet, will this have any influences on the results?
Reply: We have been also paying attention for DPAn-3 function as published in the previous paper (J Oleo Sci., 66:1149, 2017). However, it is difficult to discuss health benefits of DPAn-3 in this study, because of low fatty acid composition in tissue total lipids compared to EPA and DHA.
For the lipid class analyses and cholesterol analyses, no references were cited. How accurate is the method as compared to traditional HPLC or HPTLC and GC?
Reply: We described analytical methods in “2.1. Preparation of n-3 PUFA-PG” and “2.4. Tissue Lipid Analysis” in the text.
It would be better to provide the sum of the fatty acid, or total lipid content in the tables showing fatty acid composition, to have an idea whether they differ in the total amount.
Reply: Thank you for your advice. We added tissue lipid content in Tables to know the total amount of fatty acids.

Reviewer 3 Report
The Authors have submitted a commendable paper examining the bioavailability of n-3 PUFAs from n-3 PUFA-PG after its digestion and absorption in Diabetic/Obese KK-A y Mice. The work is well described and carefully performed. Indeed numerous organs appear collected beyond those processed for the reported assay work. However, my concern is that it is notable that no data is provided from the heart tissues, even though weights are provided. Why was heart not assayed and analysed? In light of the importance of n-3 in cardiovascular tissues and in cardiac health, especially in obesity and diabetes, this data would be greatly anticipated. The manuscript importance would increase with a report of accumulated n-3 PUFA levels in the heart. In addition, brief inclusion in the discussion regarding the importance of the mouse model itself and its prior validation for diabetes and obesity would assist the reader. Some discussion regarding limitations of mouse models with regard to dietary design and metabolic requirements are also helpful.
Author Response
Reviewer 3
The Authors have submitted a commendable paper examining the bioavailability of n-3 PUFAs from n-3 PUFA-PG after its digestion and absorption in Diabetic/Obese KK-A y Mice. The work is well described and carefully performed. Indeed numerous organs appear collected beyond those processed for the reported assay work. However, my concern is that it is notable that no data is provided from the heart tissues, even though weights are provided. Why was heart not assayed and analysed? In light of the importance of n-3 in cardiovascular tissues and in cardiac health, especially in obesity and diabetes, this data would be greatly anticipated. The manuscript importance would increase with a report of accumulated n-3 PUFA levels in the heart.
Reply: The present study focused on the effect of n-3 PUFA-PG upon diabetic/obese model mice. We will investigate heart and metabolic index for the prevention of cardiovascular diseases in further research. Thank you for your kind suggestion.
In addition, brief inclusion in the discussion regarding the importance of the mouse model itself and its prior validation for diabetes and obesity would assist the reader.
Reply: Thank you for your comment. We added “Since obesity is recognized as worldwide problem because of risk factor for diabetes, hypertension and dyslipidemia to develop metabolic syndrome, we therefore examined the effect of n-3 PUFA-PG on white adipose tissue’s weight and blood glucose level, as well as the serum and liver lipid levels of type 2 diabetic/obesity model KK-Ay mice to explore its preventive and alleviative activities.” at P10, Line14-18 in the text.
Some discussion regarding limitations of mouse models with regard to dietary design and metabolic requirements are also helpful.
Reply: Thank you for your important comment. As the next step of our research, we think clinical examination is important.

Round 2
Reviewer 1 Report
I don't have any further questions.
Reviewer 2 Report
I don't have further commentsReviewer 3 Report
Ideally if the data is on hand for the heart tissue analyses these should appear as part of this document, not as a separate study.